# Dual-Functional Monomer MIPs and Their Comparison to Mono-Functional Monomer MIPs for SPE and as Sensors

**DOI:** 10.3390/polym14173498

**Published:** 2022-08-26

**Authors:** Angela Alysia Elaine, Steven Imanuel Krisyanto, Aliya Nur Hasanah

**Affiliations:** 1Department of Pharmaceutical Analysis and Medicinal Chemistry, Faculty of Pharmacy, Universitas Padjadjaran, Jl. Raya Bandung Sumedang KM 21.5, Sumedang 45363, Indonesia; 2Drug Development Study Center, Faculty of Pharmacy, Universitas Padjadjaran, Jl. Raya Bandung Sumedang KM 21.5, Sumedang 45363, Indonesia

**Keywords:** MIP, mono-functional monomer MIP, dual-functional monomer MIP, sorbent, MI-SPE, biochemical sensor

## Abstract

A molecularly imprinted polymer (MIP) is a synthetic polymer that has characteristics such as natural receptors which are able to interact and bind to a specific molecule that is used as a template in the MIP polymerization process. MIPs have been widely developed because of the need for more selective, effective, and efficient methods for sample preparation, identification, isolation, and separation. The MIP compositions consist of a template, monomer, crosslinker, initiator, and porogenic solvent. Generally, MIPs are only synthesized using one type of monomer (mono-functional monomer); however, along with the development of MIPs, MIPs began to be synthesized using two types of monomers to improve the performance of MIPs. MIPs used for identification, separation, and molecular analysis have the most applications in solid-phase extraction (SPE) and as biochemical sensors. Until now, no review article has discussed the various studies carried out in recent years in relation to the synthesis of dual-functional monomer MIPs. This review is necessary, as an improvement in the performance of MIPs still needs to be explored, and a dual-functional monomer strategy is one way of overcoming the current performance limitations. In this review article, we discuss the techniques commonly used in the synthesis of dual-functional monomer MIPs, and the use of dual-functional monomer MIPs as sorbents in the MI-SPE method and as detection elements in biochemical sensors. The application of dual-functional monomer MIPs showed better selectivity and adsorption capacity in these areas when compared to mono-functional monomer MIPs. However, the combination of functional monomers must be selected properly, in order to achieve an effective synergistic effect and produce the ideal MIP characteristics. Therefore, studies regarding the synergistic effect of the MIP combination still need to be carried out to obtain MIPs with superior characteristics.

## 1. Introduction

Technological developments encourage the development of various methods for sample preparation and analysis. The development of many analytical and sample preparation techniques aims to increase analytical efficiency, selectivity, and repeatability [1]. One of the techniques currently being developed is a polymer medium known as a molecularly imprinted polymer (MIP), which is synthesized from monomers and characterized by its ability to absorb specific molecules. An MIP is a highly selective adsorbent polymer synthesized through the polymerization of functional monomers [1,2,3]. In the MIP synthesis process, the basic compositions are always the same, consisting of functional monomers, crosslinkers, porogenic solvent, and template molecules [1,4]. Since the imprinting technique was introduced in 1972 [5,6], MIPs have been widely applied in many fields, such as sample preparation and analysis (e.g., for extraction [7,8] and chromatography [9]), environmental monitoring [10], drug delivery [11], medical [10] and sensors [12,13,14,15]. The advantages of MIPs as a sorbent in the SPE and HPLC methods include that they are fast, low-cost, widely used to identify more compounds, can detect and separate specific molecules, high durability, and reusability. In addition, the advantages of MIPs when used as a detection element in a sensing device include being able to provide a more sensitive response, having a good sensor memory capacity, high reproducibility and stability, and low cost. Since they have many advantages, MIPs are widely used in the fields of preparation, analysis, and detection of compounds [16,17].

Several MIP synthesis methods have been developed, including emulsion polymerization, suspension polymerization [17], precipitation polymerization, bulk polymerization [18,19], electro-polymerization [20], solid-phase techniques [21], and sol-gel imprinting [22]. The selection of template and monomer molecules is critical in determining the specificity and properties of the MIP. Several types of approaches such as combinatorial methods and computational screening methods were carried out to assist the process of selecting templates and monomers [23]. The template and monomer molecules usually complement each other [1]. Generally, only one functional monomer is used in the MIP synthesis process; to date, the MIPs most widely used have one functional monomer (mono-functional monomer MIPs). Several examples of mono-functional monomer MIPs have been successfully synthesized using various methods. A mono-functional monomer MIP with trifluoperazine [24], L-phenylalanine [25], and lead ions [26] as a template was synthesized using the bulk polymerization method. The precipitation polymerization method was used to synthesize an MIP with tetracycline [27], dextromethorphan [28], and copper ions [29] as template molecules [30,31,32].

The first study on MIPs with dual functional monomers by Ramstrbm et al. succeeded in showing that MIPs synthesized with two types of functional monomers gave better performance than MIPs synthesized with a single monomer, due to the synergistic and complementary effects of the monomers [32]. These results encourage researchers to continue to develop dual-functional MIP monomers to increase the selectivity and adsorption capacity of MIPs [30,33].

The development of dual-functional monomer MIPs has led to advances in their use as a sorbent in the SPE method and as detection elements in sensor devices [15,30,33,34]. There is still a lot of research and development on dual-functional monomer MIPs to provide better selectivity and improve MIP performance. Until now, no review article has discussed the various studies relating to the synthesis of dual-functional monomer MIPs that have been carried out in recent years. This review is necessary, as an improvement in the performance of MIPs still needs to be explored and a dual-functional monomer strategy is one way to overcome the problems associated with mono-functional monomer MIPs. In this review article, we discuss the techniques commonly used in the synthesis of dual-functional monomer MIPs, and the use of dual-functional monomer MIPs as sorbents in the MI-SPE method and as detection elements in biochemical sensors.

## 2. Dual-Functional Monomer Molecularly Imprinted Solid-Phase Extraction (MI-SPE)

### 2.1. Polymerization Techniques of Dual-Functional Monomer MIPs as an MI-SPE Sorbent

The sample preparation process is an important step in the analytical process to purify and concentrate analytes in order to increase sensitivity, selectivity, and accuracy [35,36]. Currently, the sample preparation method that is widely used is solid-phase extraction (SPE), which uses a sorbent to adsorb and separate analytes from the sample [37]. SPE is widely used because it has several advantages: easy to perform, less time required, high selectivity, high efficiency, consumes fewer solvents, has the potential to be automated, and has good compatibility if combined with other analytical methods, such as the chromatography method [37,38]. In SPE, the choice of the sorbent is the critical key in determining the quality of the performance and the results of sample pre-treatment. Sorbents that are widely used in SPE, such as silica and C18, have limitations in their selectivity and specificity, resulting in poor separation results. Therefore, a sorbent with better selectivity and specificity is needed, compared with the commercially available sorbents [39,40,41,42]. Modifications to existing sorbents have been carried out for a long time to increase the specificity and selectivity of the sorbents. However, apart from that, MIPs are also an alternative sorbent in SPE [43]. MIPs are polymers with high selectivity, making them widely used in SPE as a sorbent, which is commonly known as molecularly imprinted solid-phase extraction (MI-SPE). MIPs synthesized using a template molecule that is the same or similar to the analyte in the matrix show specific recognition, binding, and isolation sites for the analyte. Hence, they can adsorb and separate the analyte better than other sorbents [2,42,44]. Currently, the polymerization methods that are widely used for the synthesis of dual-functional monomer MIPs for SPE are precipitation polymerization and bulk polymerization (Figure 1). These two polymerization methods have been widely used for the synthesis of MIPs compared to other polymerization methods because the process is simple, easy to carry out, and does not require special instruments. Briefly, dual-functional monomer MIPs that are used as sorbent in SPE are synthesized by mixing a dual-functional monomer, crosslinker, and initiator in a porogen solvent so that the polymerization process can occur. The polymer particles that form are crushed and sieved when using the bulk polymerization technique. The polymer is then eluted to completely remove the template molecule from the MIP (Figure 2).

Precipitation polymerization is a polymerization process without the use of surfactants and is initiated in a homogeneous solution containing a monomer, crosslinker, initiator, and porogenic solvent, which then produces a polymer that precipitates [45,46]. An example of a dual-functional monomer MIP as an SPE sorbent which was synthesized using the precipitation polymerization method was carried out by Wan et al. The result of the study showed that MIPs synthesized using a methanol–acetonitrile mixture (1:2, *v*/*v*) as a porogenic solvent showed a high adsorption capacity of myricetin (Q_max_ = 1.52 mg/g). The addition of glycidyl methacrylate as a co-monomer resulted in a stable polymer with high adsorption capacity and selectivity (Q_max_ = 11.8 mg/g) [34].

Nguyen et al. [10] and Zhao et al. [33] also synthesized dual-functional monomer MIPs for SPE using the precipitation polymerization method. Nguyen et al. synthesized a dual-functional monomer MIP using ciprofloxacin as a template and a combination of MAA and 2-vinylpiridine as a functional monomer. The study result stated that protic polar porogenic solvents could reduce the selectivity of the produced polymer because the solvent molecules compete with the MAA to interact with the template molecules. An aprotic polar porogen solvent is suitable for preparing MIPs as an SPE sorbent by precipitation polymerization. MIPs synthesized using a polar protic solvent (methanol) showed a low adsorption capacity (Q_max_ = 3.23 mg/g) and selectivity (imprinting factor (IF) = 0.67) when compared to MIP synthesized using an aprotic polar solvent (acetonitrile) (Q_max_ = 11.76 mg/g, IF = 1.34). The MIP synthesized with dual-functional monomer and an aprotic polar solvent showed the best performance, indicated by adsorption capacity (Q_max_ = 10.28 mg/g) and the highest IF value of the dual-functional monomer MIPs (IF = 2.55) [10]. The study by Zhao et al. synthesized a dual-functional MIP to detect polysaccharides in aqueous media by a surface imprinting technique commonly used for macromolecular recognition. Epoxy-modified silica nanoparticles were used as the solid support in the MIP synthesis by Zhao et al., which were then coated with a thin film layer in the imprinting process. Based on the Q_max_ and IF, the use of dual-functional monomers in the synthesis of MIPs as an SPE sorbent was able to increase the selectivity for the recognition of polysaccharides [33].

Bulk polymerization, such as the suspension polymerization technique, is widely used because it takes less time, is simple, requires no instruments, and uses fewer solvents. It is carried out by dissolving the template, monomer, crosslinker, and initiator in a suitable solvent [47]. However, bulk polymerization has several disadvantages, such as the need for a grinding process, the irregular shape of the resulting polymer, and the fact that the binding site can be destroyed [48]. The first synthesis of a dual-functional monomer MIP carried out by Ramstrbm et al. used the bulk polymerization technique. The dual-functional monomer MIP synthesized by Ramstrbm using a combination of 2-vinylpyridine (2VP) and MAA as monomer showed a higher selectivity factor (α 2VP-MAA = 4.35) when compared to the MIP synthesized using a single monomer (α 2VP = 2.35 and α MAA = 1.9) [32]. Another example of a dual-functional monomer MIP synthesized using the bulk polymerization method was carried out by Tang et al. Tang et al. synthesized a dual-functional monomer MIP with glycyrrhizic acid as a template compound, using β-cyclodextrin and MAA as functional monomers. The MIP made with a 1:1:10:40 molar ratio of β-cyclodextrin, glycyrrhizic acid, MAA, and crosslinker showed maximum IF results (3.77) and optimum selectivity to glycyrrhizic acid [49]. Thach et al. also synthesized a dual-functional monomer MIP using the bulk polymerization method. The dual-functional monomer MIP was synthesized with ciprofloxacin as a template, and MAA and 2VP as monomers. The percent recovery of extraction from the MIP synthesized with a dual-functional monomer and an aprotic polar solvent showed high yields of around 63–105% [50]. A summary of the results of dual-functional monomer MIP synthesis using precipitation polymerization and bulk polymerization can be seen in Table 1.

### 2.2. Dual-Functional Monomer MIP Application as an SPE Sorbent

Prior to the development of dual-functional monomer MIPs as SPE sorbents, mono-functional monomer MIPs were widely applied in MI-SPE for the analysis of many analyte compounds in food samples, biological samples, and environmental samples [51,52,53,54]. MIPs for SPE sorbents have been widely synthesized for various template compounds in the analysis of food samples, such as coumarin in chamomile tea and cinnamon samples [55], citrinin in dietary supplements and cereal samples [56], and melamine and dicyandiamide in diary product samples [57]. MI-SPE has been used in biological samples for analytes, such as tryptamine in human cerebrospinal fluid samples [58], dopamine in human urine [59], and hydroxylated metabolites of polycyclic aromatic hydrocarbons (PAHs) from human urine [60], while, it has also been used in environmental samples for analytes, such as propylparaben in wastewater and personal care products [61], imidazole fungicide in river water [62], and neurotoxin gonyautoxin 1,4 in seawater [63].

The effectiveness of the MIP is indicated by the presence of a specific binding site to the template, which can be measured from the Q_max_ and IF parameters. The synthesized polymer particles are packed into an SPE cartridge for use as a sorbent. The SPE column needs to be optimized prior to use in the sample extraction and pre-treatment process, to optimize the performance of the MI-SPE. Although MI-SPE with a single monomer has been developed, the limitations of using a single monomer result in a limitation in the performance of the MI-SPE. Therefore, many studies have developed MI-SPE using dual-functional monomers to improve performance and selectivity, and several studies have been carried out regarding the use and performance of dual-functional monomer MIPs as adsorbents in MI-SPE.

Dual-Functional Monomer MIP for Boc-L-tryptophan

The first study related to the use of two functional monomers in the MIP synthesis process was carried out by Ramstrbm et al. in 1993, who synthesized an MIP using a combination of MAA and 2VP as functional monomers for use as adsorbents for high-performance liquid chromatography (HPLC) columns for the enantiomeric separation process [32]. The study was conducted by comparing the selectivity factors of MIPs synthesized using 2VP (2VP-MIP), MAA (MAA-MIP), and a combination of 2VP and MAA (2VP-MAA-MIP) as functional monomers [32]. MAA is part of a group of acidic monomers universally used in MIP synthesis [2] and is widely used because of its hydrogen bonding donor and acceptor characteristics. MAA dimerization during MIP synthesis also enhanced the imprinting effect of the MIP. In addition, the binding capacity of the MIP can be increased by increasing the molar fraction of MAA so that the pore size in the MIP is larger and the binding capacity is increased [64,65]. The template molecule used by Ramstrbm et al. was Boc-L-tryptophan. The three types of MIP were synthesized and then applied as adsorbents in the HPLC column. The chromatographic data showed that the 2VP-MIP (IF = 2.35, see Table 1) had a better selectivity factor than the MAA-MIP (IF = 1.90), indicating the improved ability of the 2VP-MIP to separate enantiomeric compounds. However, the selectivity factor of the 2VP-MAA-MIP (IF = 4.35) was higher than the MIP synthesized with 2VP and MAA as mono-functional monomers. This study found that the use of a combination of two functional monomers in the synthesis of MIPs increased the selectivity and allowed the adsorption capacity (Q_max_) of the MIP to be improved. MAA increases the hydrophilicity and affinity of the MIP through hydrogen interactions; meanwhile, 2VP increases the affinity of the MIP through hydrophobic interactions and increases the π–π stacking interactions with the template molecules. The synergistic effect of MAA and 2VP increases the selectivity of the 2VP-MAA-MIP [34,50]. Therefore, MIPs with two functional monomers have been widely developed to synthesize with other template molecules or monomer combinations.

b.Dual-Functional Monomer MI-SPE for Ciprofloxacin and Sarafloxacin Antibiotics

The combination of 2VP and MAA as dual-functional monomers was also developed by Thach et al. and Nguyen et al. to separate ciprofloxacin (CIP) from biological or environmental samples [10,50]. CIP is a fluoroquinolone antibiotic commonly used to treat infectious diseases in humans and animals. It can accumulate in the environment because only about 30% of CIP can be metabolized in the body [66]. Many CIP-based MIPs (CIP-MIPs) have been synthesized prior to the synthesis of dual-functional monomer MI-SPE. A CIP-MIP was first synthesized in 2006 using MAA as a monomer [67]. Many studies then synthesized CIP-MIPs using different monomers, such as 2VP [68], 4-vinylbenzoic acid [69], and 1-vinyl-3-ethylimidazolium bromide [70]. However, based on the studies related to MIP synthesis with dual-functional monomers, which showed better molecular recognition and higher adsorption capacity, CIP-MIPs were synthesized using two functional monomers. Thach et al. synthesized three types of MIP, that is, 2VP-MIP, MAA-MIP, and 2VP-MAA-MIP, using the bulk polymerization method with chloroform and methanol as porogenic solvents. This study showed that MIPs synthesized using a polar aprotic porogenic solvent showed better recovery than MIPs synthesized using a weakly polar porogenic and protic polar solvent. Weakly polar solvents provide unsuitable interactions for the monomer and template, whereas polar protic solvents cause competitive interactions with functional monomers to the template molecule [71]. The CIP-MIP prepared using the single functional monomer 2VP (2VP-MIP) showed weak interaction with CIP molecules and had a low adsorption capacity. The CIP-MIP synthesized using the single monomer MAA (MAA-MIP) showed higher affinity but was not selective for CIP, indicated by a low IF. The hydrogen bond between MAA and CIP can be disrupted by the presence of polar solvent molecules such as methanol and water, making MAA non-selective towards CIP. The CIP-MIP synthesized using a dual-functional monomer combination of 2VP and MAA (2VP-MAA-MIP) showed a higher adsorption capacity (Q_max_ = 2.40 mg/g, IF = 1.66) than the 2VP-MIP (Q_max_ = 1.12 mg/g, IF = 1.51) and MAA-MIP (Q_max_ = 1.6 mg/g, IF = 0.8), proving that 2VP-MAA-MIP had higher selectivity for CIP (see Q_max_ and IF values in Table 1). This indicates that the hydrogen bond and π–π stacking of MAA and 2VP on CIP have an important role in increasing the adsorption performance of the MIP. The synthesized MIP was used to separate CIP using the SPE method. The MAA-MIP showed the lowest extraction recovery (23%), 2VP-MIP showed low extraction recovery (43–58%), and 2VP-MAA-MIP showed the highest extraction recovery (105%). The study by Thach et al. showed that the synergistic effect of the two monomers, MAA and 2VP, improved the selectivity of the MIP by forming specific adsorption sites on the MIP [72,73]. The study conducted by Nguyen et al. also showed the same result. MIP synthesized using MAA as a single monomer showed no selectivity to CIP molecules, as indicated by the low IF value (see Table 1, Q_max_ = 4.66 mg/g, IF = 0.92). The complementary interactions of 2VP and CIP enhance the selectivity of the MIP through hydrogen bonding, electrostatic interactions, and π–π stacking, so it is used as a co-monomer of MAA for the synthesis of MAA-2VP-MIP. The dual-functional monomer MIP synthesized from MAA and 2VP showed a better adsorption capacity and selectivity than MAA-MIP, as shown by the results of Q_max_ = 10.28 mg/g and IF = 2.55 (Table 1) [10,50].

The use of dual-functional monomer MIPs for the separation of antibiotics was not only developed for ciprofloxacin, but also for the separation of sarafloxacin (SAR), which is a fluoroquinolone antibiotic widely used in animal food as a veterinary medicine [74,75]. An MIP for SAR was developed in the form of restricted access media molecularly imprinted polymers (RAM-MIPs) by Cai et al. [76]. RAM-MIPs are adsorbents with a hydrophilic outer surface and a recognition site on the inner surface. They are widely used to separate sample mixtures that contain a lot of protein and other macromolecules and have a better performance than traditional MIPs in terms of the sample preparation process [77,78]. An MIP for a RAM-MIP was synthesized using 4-vinylpyridine (4VP) and MAA as functional monomers. SAR has carboxy and piperazinyl functional groups, which can form hydrogen bond donors. It can be speculated that MIPs using MAA as the single monomer have a better adsorption capacity, as can be seen in Table 1 (Q_max_ MAA-MIP = 48.56 mg/g), when compared to the MIP using 4VP as the single monomer (Q_max_ 4VP-MIP = 51.64 mg/g). However, the MIP prepared using a dual-functional monomer showed better adsorption capacity than the MIP with a single monomer (Q_max_ 4VP-MAA-MIP = 52.80 mg/g). The IF of the 4VP-MAA-MIP (5.52) also showed better results than the 4VP-MIP (IF = 4.92) and MAA-MIP (IF = 5.40), which proved that the combination of functional monomers was able to increase the selectivity and performance of the MIP, as indicated by the recovery value of the 4VP-MAA-MIP for SAR analysis in egg samples (94.0–101.3%) [76].

c.Dual-Functional Monomer MI-SPE for Natural Compound Myricetin and Glycyrrhizic Acid

The specific characteristic of dual-functional monomer MIPs has attracted attention in the separation of chemical constituents of herbal plants, such as myricetin and glycyrrhizic acid [34,49]. Myricetin is a flavonoid compound found in various herbal plants and is the active constituent of safflower and *Abelmoschus manihot* [79]. Many studies have been carried out on myricetin because of its many pharmacological activities, such as anti-platelet, anti-inflammatory, and anti-cancer [80,81,82]. However, myricetin is only available in low concentrations in herbal plants and is therefore difficult to purify and separate. Zhong et al. carried out a separation and purification method for myricetin using an MIP with modified silica as a polymer former. The IF of the synthesized MIP was 2.10, indicating the selectivity of the MIP for myricetin in the complex matrix of *Ampelopsis grossedentata* extract [83]. In addition, an MIP for myricetin was also synthesized by Xiao et al. using silica microspheres as polymer-forming compounds, giving an IF of 2.0, which was smaller than the IF of the MIP synthesized by Zhong et al. [84]. Wan et al., therefore, utilized the specific characteristic of the dual-functional monomer MIP to develop an effective method for separating and purifying myricetin from herbal plants [34], combining the monomers 4-vinylpiridine (4VP) and glycidyl methacrylate (GMA) through the precipitation polymerization method. The monomer and crosslinker used determines the stability of the resulting MIP structure. MIPs synthesized using GMA and ethylene glycol dimethacrylate (EGDMA) as a crosslinker can stabilize the MIP structure [85,86]. GMA has also attracted much attention as a monomer because it is able to provide active groups to interact with template molecules [87]. Wan et al. synthesized an MIP using a single functional monomer (4VP-MIP) and dual-functional monomers (4VP-GMA-MIP). Based on the Q_max_, the 4VP-GMA-MIP (Q_max_ = 11.8 mg/g) showed a better adsorption ability of myricetin than the 4VP-MIP (Q_max_ = 3.42 mg/g). The 4VP-GMA-MIP also had an IF of 4.9, where a high IF indicates good selectivity towards myricetin as a template molecule (see Q_max_ and IF values in Table 1) [34]. The MIP synthesized by Wan et al. was then applied as an MI-SPE sorbent to extract myricetin from safflower and *A. manihot* extract samples. The separation results were then analyzed using HPLC, giving an analytical recovery of 79.82–83.91% for the safflower sample and 81.50–84.32% for the *A. manihot* sample (Table 1). The high percent recovery shows that the ability of MI-SPE using a dual-functional monomer MIP sorbent to separate myricetin in a complex sample matrix is excellent and has the potential to be used as a method for separating natural compounds from herbal plant samples [34].

Glycyrrhizic acid is also a plant compound with many pharmacological activities, such as antiviral, anti-inflammatory, antitussive, and gastrointestinal protective agents [88,89]. The purification and enrichment of the glycyrrhizic acid content in the extraction process is therefore a concern, so a separation method with high selectivity, such as MI-SPE, has been developed. Tang et al. synthesized a dual-functional monomer MIP using MAA and β-cyclodextrin (β-CD) as functional monomers (β-CD-MAA-MIP) [49]. The MIP prepared using MAA (MAA-MIP) can be disturbed in the presence of water molecules. β-CD and its derivatives have hydrophobic and hydrophilic properties, so they are widely developed as monomers and are used to overcome the limitations of MAA, so that the synthesized MIP can separate glycyrrhizic acid in aqueous media. The MIP synthesized using a combination of β-CD and MAA showed the highest IF value (IF β-CD-MAA-MIP = 3.77) compared to the MIP synthesized with a single monomer (IF β-CD = 1.24 and IF MAA = 1.41). The synthesized dual-functional monomer MIP was then applied as an MI-SPE sorbent for glycyrrhizic acid extraction, and the extraction results were analyzed using HPLC. An increased amount of glycyrrhizic acid was extracted, and impurities were also reduced. The amount of glycyrrhizic acid that can be quantified if the analyte separation does not use an MIP as a sorbent in SPE (MI-SPE) is only about 15.03%. Meanwhile, the amount of glycyrrhizic acid that can be quantified after separation with MI-SPE is 53.63%. These results indicate that the use of an MIP as a sorbent in SPE for the separation process is able to separate more analytes from the sample. The recovery percent obtained from the analysis of glycyrrhizic acid was around 71.5–77.5%, indicating that MI-SPE is able to increase the amount recovered and has good purification properties in the separation of glycyrrhizic acid [49].

d.Dual-Functional Monomer MI-SPE for Starch

Aside from separating small molecules, dual-functional monomer MIPs can also be applied to separate macromolecules such as polysaccharides. Polysaccharides are macromolecules that play an important role in the growth and development of organisms and are known for several clinical uses, such as reducing blood lipid levels and anti-tumour and antiviral activities [89,90,91,92]. Polysaccharides have a complex structure, so the process of separating and purifying them is still an obstacle in polysaccharide research activities. Currently, polysaccharide separation and purification methods, such as precipitation, filtration, and ion-exchange chromatography, have limitations [33]. Therefore, an MIP method was developed to separate and purify polysaccharides. Previously, an MIP was developed using a single functional monomer (MAA) to separate polysaccharides [33,93]; however, the dual-functional monomer MIP showed a better adsorption capacity and imprinting effect and was therefore developed for polysaccharide separation by Zhao et al. [33]. Zhao et al. used 2-acrylamide-2-methylpropanesulphonic acid (AMPA) and 3-aminobenzeneboronic acid (APBA) as dual-functional monomers. AMPA and APBA are able to interact with one of the polysaccharide molecules, namely, starch, by covalent and non-covalent interactions. MIPs were synthesized using single monomers (AMPA-MIP and APBA-MIP) and a combination of both monomers (AMPA-APBA-MIP). The Q_max_ and IF (see Table 1) were compared to determine the selectivity of the synthesized MIP. The Q_max_ of the AMPA-MIP (4.62 mg/g) and APBA-MIP (8.89 mg/g) were lower than that of the AMPA-APBA-MIP (13.08 mg/g). The IF of the dual-functional monomer MIP (IF AMPA-APBA-MIP = 2.22) showed a higher result than the MIP synthesized with a single monomer (IF AMPA-MIP = 1.39; IF APBA-MIP = 1.63), which proved that dual-functional monomers could improve the selectivity and adsorption capacity of the MIP. The synthesized AMPA-APBA-MIP was then applied to separate and analyze starch from pine plants in aqueous media. The AMPA-APBA-MIP was able to separate starch quickly and purify polysaccharides with better performance, indicated by a comparison of the retention time of the analyzed starch and the starch used as molecular templates in the synthesis of the MIP [33].

e.Dual-Functional Monomer MI-SPE for Environmental Samples

Besides being used for analyte separation in biological samples, MIPs can also be applied to environmental samples. Xu et al. synthesized a dual-functional monomer MIP to separate methyl parathion (MP) from environmental samples. MP is a widely used insecticide that harms the environment. MP and its metabolites have low solubility, so MP can accumulate in water and soil. Thus, the determination of the amount and separation of MP becomes important [94,95,96]. However, several methods for the detection and separation of MP, such as HPLC and gas chromatography (GC), are not capable of detecting MP in complex sample matrices due to the low detectability of these methods [96]. The studies that have been carried out demonstrate that MIP has good selectivity, and the use of a dual-functional monomer can increase the selectivity and adsorption capacity of the MIP. Therefore, MIPs are used as a method of separating MP in environmental samples. Xu et al. synthesized an MIP using 4VP and MAA as a combination of functional monomers, as they can form hydrogen bonds and π–π stacking on the MIP, to increase the interaction between the analyte and the MIP. The effectiveness of the MIP is indicated by the presence of a specific binding site to the template, which can be measured from the Q_max_ and IF parameters. The Q_max_ of the MAA-MIP (3.5 mg/g) was greater than that of the 4VP-MIP (1.25 mg/g) and 4VP-MAA-MIP (2 mg/g), indicating that increasing the amount of MAA in the synthesized MIP also increases the adsorption capacity. Meanwhile, the IF of the 4VP-MIP (4.4) was higher than that of the MAA-MIP (1.4), indicating that increasing the amount of 4VP could increase the IF; however, the combination of MAA and 4VP as functional monomers resulted in a higher IF (5). The MIP was then applied to the pre-concentration process to determine the amount of MP in the soil sample; the recovery value from the analysis using the MI-SPE method was 81.1–87.0% [96]. A summary of the dual-functional applications for MI-SPE and the results described above is provided in Table 1.

**Table 1 polymers-14-03498-t001:** Dual-Functional Monomer MIP Applications as an MI-SPE Adsorbent.

No	Monomer	Template Molecule	Q_max_ (mg/g)		IF		% Recovery	Ref
			SM	DM	SM	DM		
**1**	MAA and 2VP	Boc-L-Tryptophan	0.115 (2VP)	0.058	2.35 (2VP)	4.35	-	[32]
			0.035 (MAA)		1.9 (MAA)			
**2**	MAA and 2VP	Ciprofloxacin	1.12 (2VP)	2.4	1.51 (2VP)	1.66	105%	[50]
			1.6 (MAA)		0.8 (MAA)			
**3**	MAA and 2VP	Ciprofloxacin	4.66 (MAA)	10.28	0.92 (MAA)	2.55	65.97–119.26%	[10]
**4**	MAA and 4VP	Sarafloxacin	51.64 (4VP)	52.80	4.94 (4VP)	5.52	94.0–101.3%	[76]
			48.56 (MAA)		5.4 (MAA)			
**5**	4VP and GMA	Myricetin	3.42 (4VP)	11.8	-	4.9	79.82–84.32%	[34]
**6**	β-CD and MAA	Glycyrrhizic Acid	75.4 (β-CD)	69.3	1.24 (β-CD)	3.77	71.5–77.5%	[49]
			69.1 (MAA)		1.41 (MAA)			
**7**	APBA and AMPA	Polysaccharide (starch)	8.89 (APBA)	13.08	1.63 (APBA)	2.22	-	[33]
			4.62 (AMPA)		1.39 (AMPA)			
**8**	MAA and 4VP	Methyl parathion	1.25 (4VP)	2	4.4 (4VP)	5	81.1–87.0%	[96]
			3.5 (MAA)		1.4 (MAA)			

## 3. Dual-Functional Monomer MIPs as Detection Elements in Sensors

### 3.1. Polymerization Techniques of Dual-Functional Monomer MIPs as Recognition Elements in Sensors

In order to polymerize dual-functional monomers for sensing purposes, one should acknowledge whether the yielded bond between the template or monomer is covalent or non-covalent. A non-covalent bond is more versatile, since binding and unbinding multiple times is subsequently easier than for its covalent counterpart. Even though sensing MIPs can technically be synthesized for any kind of analyte, due to the absence of a specific optimization method for any particular group of analytes, each sensing MIP is unique for one analyte, and the development for another analyte may take a long time [2].

The polymerization methods commonly used for the synthesis of dual-functional MIP monomers for detection elements for sensors are bulk polymerization, electro-polymerization (Figure 3a), and emulsion polymerization (Figure 3b). Electro polymerization is a surface imprinting technique. In the in situ electro-polymerization processes, cyclic voltammetry (CV) is used to produce very thin polymer films. The polymer film production process is well controlled when using CV because it can determine and regulate the adhesion and morphology characteristics of the resulting polymer films. The use of CV can also optimize MIP synthesis so that the MIP has high reproducibility. When the template is removed from the polymer film layer, the imprinting cavity is on the surface or close to the surface of the polymer film [97,98]. In the electro-polymerization technique, the monomer is oxidized on the electrode surface. The electrode can be coated with glass, tin, gold, or platinum. Oxidation of monomers can occur by ion transfer in a solvent medium that is stable in oxidation conditions of the monomer and able to provide a conductive medium. [98,99]. Emulsion polymerization uses an oil-in-water system in which the functional monomer, crosslinker, and template are emulsified in water and a stabilizer is added. Uniform emulsion droplets may be produced, but the purity of the yielded polymer is low, and this technique is difficult due to the need to maintain the correct level of water [97,100].

### 3.2. Dual−Functional Monomer MIP Applications as Recognition Elements in Sensors

In this method of application, polymerized dual-functional monomer MIPs are used as detection elements in sensing devices for the identification of molecules of compounds. This change can yield a notable advantage, in addition to the usual benefit received from a mainstream template, notably a high absorption coefficient for visible light, which allows a lower limit of detection (LOD), enabling substance detection (especially qualitative detection) at even lower concentrations. This benefit is more distinct in compounds with a lower overall value of detection and/or concentration, for example, salicylic acid bioanalysis and most pesticide analyses [92,95]. Another benefit is high carrier mobility and small excitation binding energy, allowing the detection to run faster and reducing the LOD even further. The final benefit is the long-range excision diffusion lengths, which are a small benefit compared to the other benefits [95].

Xia et al. used 1-butyl-3-methylimidazolium 5 hexafluorophosphate activated by multi-wall carbon nanotubes called Nafion-MIP-MWCNTs@IL/GCE as sensors for L-Trp (L-tryptophan). L-Trp itself is a precursor neurotransmitter that can improve sleep and immunity, and is also applied in Parkinson’s and Pellagra medication. Overdosing causes serious problems, such as muscle pain, stomach pain, and shortness of breath. The sensor had a low LOD (6 nM) and low non-specific adsorption, as well as being stable at a pH value of 2.5. This stability decreased as the pH value shifted to base, which was reflected in a reduced peak. In terms of selectivity, the sensor was recognized as being highly selective because the peak only shifted 4–8% when facing tenfold interferences from L-tyrosin and L-phenylalanine, whose structures are similar to L-Trp. Nafion-MIP-MWCNTs@IL/GCE showed good reproducibility, as shown by its relative standard deviation (RSD) of 3.5% (*n* = 5). The sensor was also able to retain 96.3% and 92.7% of its initial response after storage for 15 days and 30 days in a refrigerator, respectively, thus confirming its good stability [100].

Aside from the chemistry and environment field, dual-functional monomer sensing can be applied to the medical field. For instance, a study by Nawaz et al. used MAA and itaconic acid as functional monomers to form a surface plasmon resonance (SPR) sensor to detect tetracycline levels in urine. Tetracycline is a widely used antibiotic for the treatment of diseases in humans and animals. SPR uses a refractive index in the nanometer range to measure analytes and is proven in synergy with MIPs. The LOD and limit of quantification (LOQ) values were 1.38 × 10^−14^ mol/L and 4.5 × 10^−14^ mol/L, respectively, when performing triple measurements in three different urine samples. The linearity was great, with a value of 0.999, and it proved potent even in complex matrices, something that the tetracycline standard failed to achieve. The sensor was also able to retain 89.14% and 80.73% of the initial refractivity index after being stored at room temperature for 30 and 60 days, respectively [101].

Hudson et al. also synthesized a sensor for antibiotic detection by the free-radical polymerization of MAA and FluMa (fluorescein methacrylate). Antimicrobial resistance is considered a global threat comparable to terrorism and climate change, with 700,000 estimated casualties per year worldwide, and is anticipated to reach 10 million with no intervention. One preventive approach is monitoring antibiotic residual levels in the environment, which can be executed by dual-functional monomer sensors. The sensor produced by Hudson et al. had a LOD value of 5 mM and could be further improved with polymer thinning. The sensitivity and selectivity were deemed to be high and it was able to perform well against multiple targets, namely beta-lactam and tetracycline groups [102].

Shen and Kan demonstrate the use of MIP sensing for the recognition and sensing of dopamine using ionic polymerization. This is mainly due to the high demand for a new method of determination of neurotransmitters with good selectivity and high sensitivity. The ion functional monomers consist of Lithium Percloratte and pyrrole, which can be enhanced using the cyclic voltammetry method. The resulting MIP possesses a higher recognition ability compared to its single counterpart. The obtained LOD is 5.0 × 10^−8^–1.0 × 10^−5^ mol/L in a linear range and, with an RSD of just 0.3%, the result is relatively acceptable and reproducible [103]. We summarize the dual-functional application of MIP monomers as sensors in Table 2.

The dual-functional monomer MIP is proven to be capable of possessing extremely low LOD and LOQ, often in the nano measurement range. The dual-functional monomer MIP provides a good detection range and stability, being able to retain most of its ability after storage for months, and good selectivity, being able to perform well, even in complex environments. Further study is needed to research more effective and efficient polymerization techniques, with minimal drawbacks to sensor quality.

**Table 2 polymers-14-03498-t002:** Dual-Functional Monomer MIP Applications as Sensors.

No.	Monomer	Template	LOD (M)	Ref
1	Styrene and 4-vinylbenzoic acid	L-Tryptophan	6 × 10^−6^	[100]
2	MAA and IA	Tetracycline	1.38 × 10^−14^	[101]
3	MAA and FluMa	Nafcillin	5 × 10^−3^	[102]

## 4. Conclusions

Based on several studies and research on dual-functional monomer MIPs, it can be stated that the use of multiple monomers in MIP synthesis has a synergistic effect on the MIP performance, so that the selectivity, adsorption capacity, and efficiency of the MIP can be improved. The factors that must be considered in the synthesis of dual-functional monomer MIPs are the same as in the synthesis of MIPs with single monomers. The superior properties of the dual-functional monomer MIPs can be achieved through an optimization process during MIP synthesis. The polymerization of dual-functional MIP monomers can be influenced by several factors, such as the type of monomers being used and combined, the number of monomers, crosslinkers, initiators, type and polarity of the solvents used in the MIP synthesis, temperature, and the duration of the polymerization reaction [55]. Therefore, the optimization process is essential to obtain the conditions in which the optimum properties of the dual-functional monomer MIP can be achieved. Using dual-functional monomer MIPs as an adsorbent in the SPE method and as a detection element for biochemical sensing can improve the separation, purification yields, and analysis process of analytes from a complex sample matrix. However, the combination of monomers used so far has not varied, so the synthesis of dual-functional monomer MIPs can still be developed with other monomer combinations that can give a synergic effect to improve the MIP performance; thus, more research, development, and exploration are needed to utilize the potential of sensing dual-functional monomer MIPs.

## Figures and Tables

**Figure 1 polymers-14-03498-f001:**
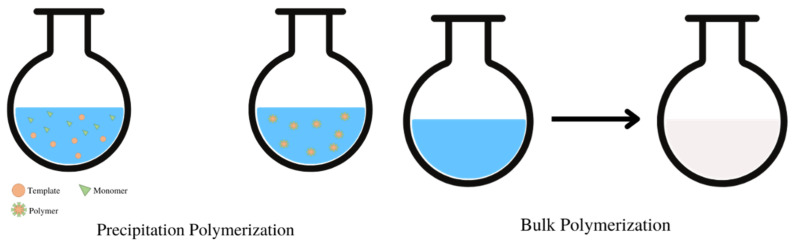
Illustration of the precipitation polymerization method (**left**) and bulk polymerization method (**right**).

**Figure 2 polymers-14-03498-f002:**
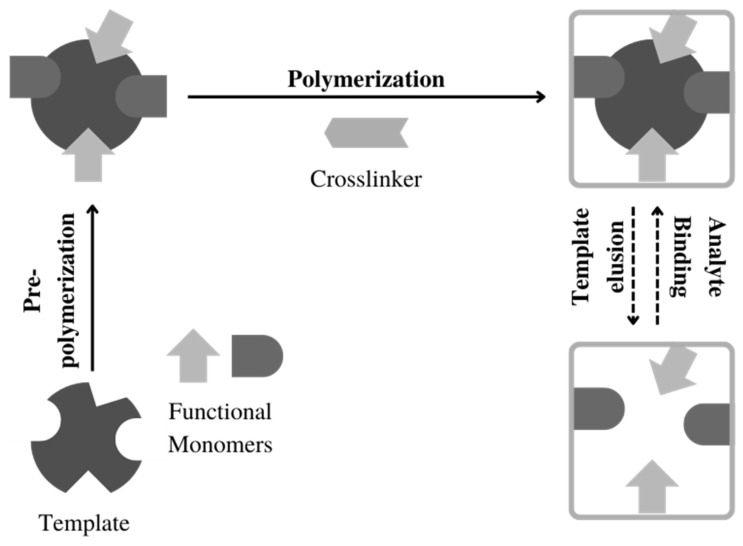
Illustration of the polymerization process of dual-functional monomer MIPs.

**Figure 3 polymers-14-03498-f003:**
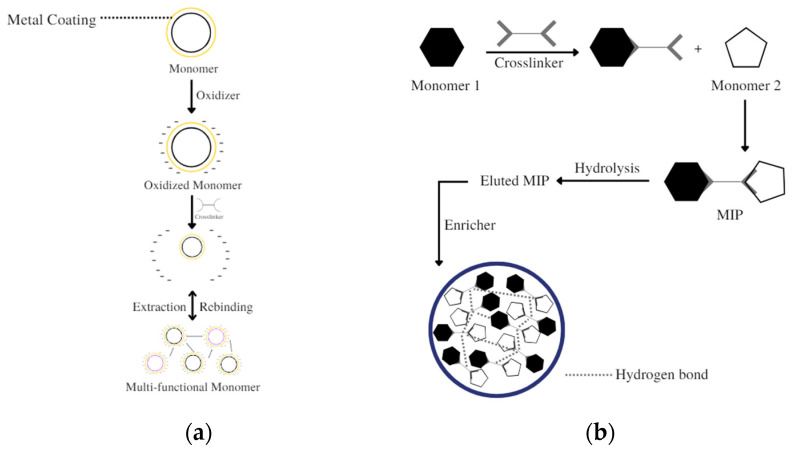
(**a**) Illustration of surface imprinting/electro−polymerization. (**b**) Illustration of emulsion polymerization.

## Data Availability

No new data generated during the study.

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
