# Peer review of "Dual-Functional Monomer MIPs and Their Comparison to Mono-Functional Monomer MIPs for SPE and as Sensors"

_polymers, 2022, doi:10.3390/polym14173498_

Round 1

Reviewer 1 Report

In this review manuscript a general overview of the imprinted polymers obtained with the use of two different monomers and their applications for SPE and sensing have been provided. The topic is interesting. Brief introduction connected with MIPs  synthesis, importance and their application in SPE techniques and sensors was described. Despite all the previously mentioned, there are different aspects that should be considered and modified before considering this manuscript for publication. Below are major (but not all) problems that could help Authors to improve their manuscript.

1.     Page 1 line 43: Authors wrote : “Since its (MIPs) development in 1994 …” – is it correct??? MIPs were developed and described in 70’s! In page 7 lines 216-217 Authors wrote that “The firs study related to the use of two functional monomers in the MIP synthesis process was carried out by Ramstrbm et al. in 1993” !!!

2.     There is lack of adequate citation connected with the MIP development (page 1, line 217). It is necessary to complete this information.

3.     In the Introduction (page 2 lines 49-73) Authors described mono-monomer MIPs (methods of synthesis) but no references were added in the text. There is the table but it is not sufficient.

4.     Table 1 refers to the mono-monomer MIPs but the manuscript should be about dual-monomer MIP. Such comprehensive table is not necessary.

5.     Section 2 title is Monomer imprinted polymer solid phase extraction – is it correct? What does it mean?

6.     The content of section 2.1 and 2.2 (and 3.1 and 3.2) is very similar. The same information, and the same publications are described twice. It is not correct. Authors should consider to combine section 2.1 and 2.2 (as well as 3.1 and 3.2) and make one section.

7.     In the description there is many mistakes, e.g. page 4 line122:functional monomer MIP (4-vinylpyridine) – what is functional monomer MIP??? MIP should be missing.

8.     The description of MIP preparation processes is too detailed. Authors should rather concentrate to show the dual MIP potential and give the critical view and future prospects.

9.     Fig .1 is not good. It does not show anything. It should be cancelled or totally changed.

10.  Page 5 lines 142-144: In line 144 Authors wrote that “…, indicated by the highest adsorption capacity (Qmax = 10.28 mg/g)” but earlier (line 142) was information that polymer made from MAA (one monomer) showed Qmax= 11.76 mg/g??!!!

11.  Page 5 line 160 – what is soxhletation???

12.  Section 2.2 is poorly written. There is not general idea of the section. Firstly the Authors described historically first publication (page 7 lines 216-238), then they focus on the type of monomers (page 8, lines 241-279), and then (page 9, lines 283-315) Authors pay main attention according to the type of template/analyte used. What is the main idea??? The section is not consistent. In page 8, lines 241-279 Authors described MIP for ciprofloxacin – antibiotic, then (page 9) we have description of MIPs dedicated to chemical compounds separated from plants and (page 10) macromolecules, and in page 11 we have again information about antibiotics – why in that way. And, moreover, (page 11, lines 385-404) Authors started to pay attention on the type of analyzed samples. What is the main idea???

13.  Page 8, lines 265-266: there are the same values of Qmax and IF connected with different MIPs.

14.  Page 9, line 291 – what is polymer former???

15.  Page 9, line 291-292, Authors wrote that when IF for MIP is 2.1 the MIP is not specific for myricetin. Is not true – when IF is more than 1 the imprinting process is successful.

16.  Page 10, lines 331-333: Authors wrote :Before the MI-SPE procedure, the amount of glycyrrhizic acid that could be analysed was only 15.03%. – What does it mean, what for is 15.03%???

17.  In page 11, lines 377-380 Authors showed Qmax values: 0.348, 0.327, 0.355 – they are very similar.

18.  In lines 414-426 Authors was similar information as in lines 222-227.

19.  Lines 453-462 should be rewritten.

20.  In lines 463-473 Authors described bulk polymerization what was earlier described in lines 156-162.

21.  In the manuscript there is lack of conclusions, future perspectives and critical comments.

22.  Additionally, some important examples of dual-monomer MIPs were omitted:

-        https://doi.org/10.1016/j.jmrt.2021.08.023

-        https://doi.org/10.1016/j.reactfunctpolym.2021.105152

-        Journal of Chromatography B, Volume 1193, 2022, 123172

-        https://doi.org/10.1016/j.foodchem.2022.132792

-        https://doi.org/10.1016/j.aca.2021.339245

-        https://doi.org/10.1002/slct.202104038

In my opinion, the manuscript in current form is not suitable for publication. I recommend major revision of the manuscript.

Author Response

In this review manuscript a general overview of the imprinted polymers obtained with the use of two different monomers and their applications for SPE and sensing have been provided. The topic is interesting. Brief introduction connected with MIPs  synthesis, importance and their application in SPE techniques and sensors was described. Despite all the previously mentioned, there are different aspects that should be considered and modified before considering this manuscript for publication. Below are major (but not all) problems that could help Authors to improve their manuscript.

  1. Page 1 line 43: Authors wrote : “Since its (MIPs) development in 1994 …” – is it correct??? MIPs were developed and described in 70’s! In page 7 lines 216-217 Authors wrote that “The firs study related to the use of two functional monomers in the MIP synthesis process was carried out by Ramstrbm et al. in 1993” !!!

Response:

Thank you for your comment. We have corrected the statement regarding the first development of MIP on Page 2 lines 52 – 57.

The correction is below :

Since the imprinting technique was introduced in 1972 [4,5], MIP has been widely applied in many fields, such as sample preparation and analysis (e.g., for extraction [6,7] and chromatography [8]), environmental monitoring [9], drug delivery [10], medical [9] and sensors[11–14]. MIPs are widely used because they are easy to synthesize, have good stability and robustness, and have high selectivity for specific molecules [15,16].

  1. There is lack of adequate citation connected with the MIP development (page 1, line 217). It is necessary to complete this information.

Response:

Thank you for your comment, we added some references connected to MIP development on page 2, line 49 and page 8, line 250.

  1. In the Introduction (page 2 lines 49-73) Authors described mono-monomer MIPs (methods of synthesis) but no references were added in the text. There is the table but it is not sufficient.

Response:

We added some references of each MIP example synthesized by the bulk polymerization (page 2, lines 76 – 78) and precipitation polymerization (page 2, lines 78 – 80) methods have been added.

  1. Table 1 refers to the mono-monomer MIPs but the manuscript should be about dual-monomer MIP. Such comprehensive table is not necessary.

Response:

Thank you for your comment, we deleted table 1.

  1. Section 2 title is Monomer imprinted polymer solid phase extraction – is it correct? What does it mean?

Response:

Thank you for your comment. The title of the section should be Dual-Functional Monomer Molecular Imprinted Solid Phase Extraction (MI-SPE) (page 4 lines 106 – 107). This section is then divided into two sub-sections, the first section is about the technique of synthesizing dual-functional monomer MIP as a sorbent in SPE, and the second section is on the application of dual-functional monomer MIP as a sorbent in SPE sorbent and its comparison with mono-functional monomer MIP as sorbent SPE.

  1. The content of sections 2.1 and 2.2 (and 3.1 and 3.2) is very similar. The same information, and the same publications are described twice. It is not correct. Authors should consider to combine section 2.1 and 2.2 (as well as 3.1 and 3.2) and make one section.

Response:

Thank you for your comment, in the revision article section 2.1 and section 2.2 are not combined, but we divided section 2.2 again into subsections according to the topic to make it easier for readers to understand the contents of the section.

  1. In the description there is many mistakes, e.g. page 4 line122:functional monomer MIP (4-vinylpyridine) – what is functional monomer MIP??? MIP should be missing.

Response:

Thank you for your comment, we have corrected it on page 4 line 141.

The correction is “functional monomer (4-vinylpyridine)

  1. The description of MIP preparation processes is too detailed. Authors should rather concentrate to show the dual MIP potential and give the critical view and future prospects.

Response:

Thank your for your comment, the MIP preparation process section has been shortened to make it easier for readers to understand  on page 4 lines 135 – 151.

  1. Fig .1 is not good. It does not show anything. It should be canceled or totally changed.

Response:

figure 1 has been fixed on page 5.

  1. Page 5 lines 142-144: In line 144 Authors wrote that “…, indicated by the highest adsorption capacity (Qmax = 10.28 mg/g)” but earlier (line 142) was information that polymer made from MAA (one monomer) showed Qmax= 11.76 mg/g??!!!

Response:

Thank your for your comment, apologize on our mistakes. A higher imprinting factor (IF) value indicates a better selectivity of MIP. The highest IF value was given by MIP, which used two types of functional monomers (IF = 2.55). Supposedly, the word "the high" refers to the value of the imprinting factor, not the adsorption capacity. We corrected it on page 5 lines 170.

  1. Page 5 line 160 – what is soxhletation???

Response:

Soxhletation refers Soxhlet method which is the method used for the extraction of template molecules  from MIP in the MIP synthesis process.  Deleted the information on that as that not necessary to write.

  1. Section 2.2 is poorly written. There is not general idea of the section. Firstly the Authors described historically first publication (page 7 lines 216-238), then they focus on the type of monomers (page 8, lines 241-279), and then (page 9, lines 283-315) Authors pay main attention according to the type of template/analyte used. What is the main idea??? The section is not consistent. In page 8, lines 241-279 Authors described MIP for ciprofloxacin – antibiotic, then (page 9) we have description of MIPs dedicated to chemical compounds separated from plants and (page 10) macromolecules, and in page 11 we have again information about antibiotics – why in that way. And, moreover, (page 11, lines 385-404) Authors started to pay attention on the type of analyzed samples. What is the main idea???

Response:

Thank you for your comment. In section 2.2., the author would like to convey about the dual-functional monomer MIP, which has been synthesized and used as a sorbent in MI-SPE. In section 2.2., we also discusses the comparison of the results of the adsorption capacity and selectivity of dual-functional monomer MIP with mono-functional monomer MIP so that it can be concluded that dual-functional monomer MIP has better adsorption capacity and selectivity than mono-functional monomer MIP. We created a subsection to make it easier for readers to know what topics are being discussed on the revised manuscript.

  1. Page 8, lines 265-266: there are the same values of Qmax and IF connected with different MIPs.

Response:

Thank your for your comment. We corrected the Qmax and IF values on page 9, lines 293 – 301).

It should be “CIP-MIP synthesized using a dual-functional monomer combination of 2VP and MAA (2VP-MAA-MIP) showed a higher adsorption capacity (Qmax = 2.40 mg/g, IF = 1.66) than 2VP-MIP (Qmax = 1.12 mg /g, IF = 1.51) and MAA-MIP (Qmax = 1.6 mg/g, IF = 0.8), proving that 2VP-MAA-MIP had higher selectivity for CIP (see Qmax and IF values in Table 2).”

  1. Page 9, line 291 – what is polymer former???

Response:

Thank your for your comment. The phrase polymer former refers to the silica which modified to form a MIP polymer that has the characteristics of recognizing the template molecule.

  1. Page 9, line 291-292, Authors wrote that when IF for MIP is 2.1 the MIP is not specific for myricetin. Is not true – when IF is more than 1 the imprinting process is successful.

Response:

Thank you for your comment, we have revised the statement and deleted “specific” word on page 9 line 316..

  1. Page 10, lines 331-333: Authors wrote :Before the MI-SPE procedure, the amount of glycyrrhizic acid that could be analysed was only 15.03%. – What does it mean, what for is 15.03%???

Response:

Thank you for your comment. We revised the statement on page 11 line 397-402. The revision is below :

Glycyrrhizic acid quantified in the analysis process is only 15.03% if the separation process uses SPE with commonly used sorbent. Meanwhile, if the separation of glycyrrhizic acid using dual-functional monomer MI-SPE as a sorbent, glycyrrhizic acid that can be quantified is about 53.63%. This indicates an increase in the amount of glycyrrhizic acid that can be separated or extracted from the sample matrix.

  1. In page 11, lines 377-380 Authors showed Qmax values: 0.348, 0.327, 0.355 – they are very similar.

Response:

Thank you for your comment. There is an error in the calculation of the data conversion. We corrected the Qmax values based on calculation on Page 9, lines 327 – 333.

  1. In lines 414-426 Authors was similar information as in lines 222-227.

Response:

Thank you for your comment. We deleted the similar information in lines 414 – 426 and completed the information in lines 252 – 257.

  1. Lines 453-462 should be rewritten.

Response:

Thank you for your comment. We have rewritten lines 453 – 462 as suggestion.

The correction is below

The polymerization methods commonly used for the synthesis of dual-functional MIP monomers for detection elements for sensors are bulk polymerization, electro-polymerization (figure 3a), and emulsion polymerization (figure 3b). Electro polymerization is a surface imprinting technique (page 15 lines 474 – 477)

  1. In lines 463-473 Authors described bulk polymerization what was earlier described in lines 156-162.

Response:

Thank you for your comment. We deleted information about bulk polymerization in lines 463 - 473 and described it on lines 156 - 162.

  1. Style of the text should be improved. The authors are also invited to refine the language of the paper to make the story flow smoothly.Grammatical mistakes should be corrected.

Response:

We have done proofreading for the submitted manuscript with proof-reading-service.com with ref. no. 202207-423305.

Reviewer 2 Report

Line 11: “Molecularly imprinted polymer (MIP) is a polymer…” should be “A molecularly….”

Line 15: “Generally, MIP is only…” should be “Generally, MIPs are only”.

Line 17: should be “MIPs with two”

Line 18: should be “MIPs used for…”

The introduction states that MIPs were developed in 1994, however they found their origins way before this point. Maybe a short sentence about the inception of molecularly imprinting prior to mentioning their further development.

Line 43: “Since its development…” should be “ Since their development in 1994, MIPs have been…”

When mentioning the different method of preparing MIPS (line 49 – 51), the authors fail to mentioned solid phase synthesis of MIPs.

Line 170: should say “higher selectivity” rather than “high selectivity”

Line 199” should says “MIPs” not “MIP”

Line 445 – 462: This paragraph doesn’t really make sense. This should be re-written. Stating that monomers are coated with silica or rust-resistant electrodes. Surely, you mean that a substrate can be coated with monomers/silica, or that electrodes can be instead? You also mentioned that glass is an electrode – glass is clearly a substrate, unless coated with a conducting layer e.g. ITO.

The whole manuscript feels very repetitive, and mentioned the same preparation methods as the introduction time and time again. Further bulk is added by consistently mentioning the benefits of each of these methods, when this stated prior in the review.

Compared to the rest of the manuscript the sensor section is massively under developed with fewer than 10 references made, with some of them even recycled from previous paragraphs.

The English throughout the manuscript needs checking over and improving.

Author Response

  1. Line 11: “Molecularly imprinted polymer (MIP) is a polymer…” should be “A molecularly….”

Response:

Thank you for your comment. We corrected the sentence as suggestion on page 1 lines 11 - 13.

The correction is below

A Molecularly imprinted polymer (MIP) is a synthetic polymer that has characteristics such as natural receptors which are able to interact and bind to a specific molecule that is used as a template in the MIP polymerization process

  1. Line 15: “Generally, MIP is only…” should be “Generally, MIPs are only”.

Response:

Thank you for your comment. We corrected the sentence on page 1 lines 17 – 19.

The correction is below

Generally, MIPs are only synthesized using one type of monomer (mono-functional monomer); but along with the development of MIP, MIP began to be synthesized using two types of monomers to improve the performance of MIP.

  1. Line 17: should be “MIPs with two”

Response:

Thank you for your comment. We have rearrange the abstract section.

  1. Line 18: should be “MIPs used for…”

Response:

Thank you for your comment. We corrected the sentence on page 1 lines 22 – 23.

The correction is below :

MIPs used for identification, separation and molecular analysis has the most applications in solid phase extraction (SPE) and as biochemical sensors.

  1. The introduction states that MIPs were developed in 1994, however they found their origins way before this point. Maybe a short sentence about the inception of molecularly imprinting prior to mentioning their further development.

Responses:

Thank you for your comment. In the introduction, information has been added regarding the first study on imprinting technology and MIP development in 1972 (page 2, lines 52 – 53) (https://doi.org/10.1002/bip.1972.360110213 and https://doi.org/10.1002/macp.1989.021900724). In addition, the text also includes information that the type of MIP developed in 1994 is a dual-functional monomer MIP (page 2 lines 83 – 84).

  1. Line 43: “Since its development…” should be “ Since their development in 1994, MIPs have been…”

Response:

Thank you for your comment. We corrected the sentence on page 2 lines 52 – 58.

The correction is below :

Since imprinting technique was introduced in 1972, MIP has been widely applied in many fields, such as sample preparation and analysis (e.g., for extraction and chromatography), environmental monitoring, drug delivery, medical and sensor.

  1. When mentioning the different method of preparing MIPS (line 49 – 51), the authors fail to mentioned solid phase synthesis of MIPs.

Response:

Thank you for your comment. Solid phase synthesis has been mentioned on page 2 line 61.

  1. Line 170: should say “higher selectivity” rather than “high selectivity”

Response:

Thank you for your comment. We corrected the sentence on page 6 lines 189 – 193.

The correction is below :

Dual-functional monomer MIP synthesized by Ramstrbm using a combination of 2-vinylpyridine (2VP) and MAA as monomer showed a higher selectivity factor (α 2VP-MAA = 4.35) when compared to MIP synthesized using a single monomer (α 2VP = 2.35 and α MAA = 1.9)

  1. Line 199” should says “MIPs” not “MIP”

Response:

Thank you for your comment. We changed “MIP” to “MIPs” on page 7 lines 225.

  1. Line 445 – 462: This paragraph doesn’t really make sense. This should be re-written. Stating that monomers are coated with silica or rust-resistant electrodes. Surely, you mean that a substrate can be coated with monomers/silica, or that electrodes can be instead? You also mentioned that glass is an electrode – glass is clearly a substrate, unless coated with a conducting layer e.g. ITO.

Response:

Thank you for your comment. We rearranged lines 445 – 462 (in revised manuscript: in lines 480 – 501).

The correction in below:

In the in situ electro-polymerization processes, cyclic voltammetry is used to produce very thin polymer films. The polymer film production process is well controlled when using cyclic voltammetry because cyclic voltammetry can determine and regulate the adhesion and morphology characteristics of the resulting polymer films. The use of cyclic voltammetry can also optimize MIP synthesis so that MIP has high reproducibility. When the template is removed from the polymer film layer, the imprinting cavity is on the surface or close to the surface of the polymer film (page 15 lines 480 – 487).

In the electro-polymerization technique, the monomer is oxidized on the electrode surface. The electrode can be coated with glass, tin, gold, or platinum. Oxidation of monomers can occur by ion transfer in a solvent medium that is stable in oxidation conditions of monomer and able to provide a conductive medium. (page 15 lines 489 – 493).

  1. Compared to the rest of the manuscript the sensor section is massively under developed with fewer than 10 references made, with some of them even recycled from previous paragraphs.

Response:

Thank you for your comment. Section 3 has been rewritten and improved. Some of the information described in the previous paragraph has been deleted.

  1. The English throughout the manuscript needs checking over and improving.

Response:

We have done proofreading for the submitted manuscript with proof-reading-service.com with ref. no. 202207-423305.

Reviewer 3 Report

The review article describes the potential of dual-monomer MIP in comparison with mono-monomer ones emphasising how the use of two monomers can improve the selectivity and, in general, the performance of MIPs especially in solid-phase extraction methods. Despite MIP technology is well known and was largely applied in the last two decades, this comparison was never presented at this level of detail in the recent literature to the best of the reviewer knowledge. The authors focused on important aspect of MIP technology providing a representative selection of examples. The considerations regarding MIP design (i.e., rational monomer design strategies), the examples of sensing application (conductive/non-conductive MIP) and other contents should be further improved before publication. Please, consider revising the following points:

1)    Include a more specific definition of MIP in the abstract (target-mimicking cavities)

2)    Consider improving the list of keywords by adding mono-monomer MIP and MI-SPE or others (up to 6 keywords are suggested, especially for a review article)

3)    Introduction (line 51): The selection of a suitable monomer (or more) for the design of a target-specific MIP is surely challenging. However, in the last decade, rational design methods have been widely applied to solve this issue. These approaches allow the computation screening of large libraries of monomers against the target/template molecule and evaluate the interactions at stake (depending on the system under study). Please consider including articles for Prof.  Sergey A. Piletsky (both reviews and examples) :  https://doi.org/10.1080/01496395.2017.1287197; 10.1039/C2AN35228A; https://doi.org/10.1016/j.snb.2019.126786; https://doi.org/10.1002/elan.201900397. If you can add a comment on the very first steps towards computational MIP design: https://doi.org/10.1002/(SICI)1099-1352(199812)11:1/6<79::AID-JMR394>3.0.CO;2-B.

4)    Introduction (line 48): When listing the different applications of MIP please provide also a brief list of the pro and cons that are related to this technology (all the advantages related to the compatibility with other materials, their stability compared to bio-based systems, the capacity to simulate key-lock enzymatic interactions, the possibility of being redesigned). In respect to this last point you can add this recent example: https://doi.org/10.3390/s19204433.

5)    Please improve the figure resolution and make the labels more readable starting from Figure 2.

6)    Section 2 (line 185 on): please consider dividing the paragraph 2.2 in subsections to help the reader going through the text and understanding better the concepts described.

7)     After section 3, please include a last section 4 for the final remarks and the conclusions. Please provide insights about the future developments of these technology and how the state-of-the-art provided in your review can inspire/contribute to the improve this research field.

Author Response

  1. Include a more specific definition of MIP in the abstract (target-mimicking cavities)

Response:

Thank you for your comment. We corrected the definition of MIP in the abstract page 1 lines 11 – 13.

The correction is below:

A Molecularly imprinted polymer (MIP) is a synthetic polymer that has characteristics such as natural receptors which are able to interact and bind to a specific molecule that is used as a template in the MIP polymerization process.

  1. Consider improving the list of keywords by adding mono-monomer MIP and MI-SPE or others (up to 6 keywords are suggested, especially for a review article)

Response:

Thank you for your comment. We added some keywords on page 1 lines 38 – 39.

Keywords: MIP; Mono-functional monomer MIP; Dual-functional monomer MIP; Sorbent; MI-SPE; Biochemical sensor.

  1. Introduction (line 51): The selection of a suitable monomer (or more) for the design of a target-specific MIP is surely challenging. However, in the last decade, rational design methods have been widely applied to solve this issue. These approaches allow the computation screening of large libraries of monomers against the target/template molecule and evaluate the interactions at stake (depending on the system under study). Please consider including articles for Prof.  Sergey A. Piletsky (both reviews and examples): https://doi.org/10.1080/01496395.2017.1287197; 10.1039/C2AN35228A; https://doi.org/10.1016/j.snb.2019.126786; https://doi.org/10.1002/elan.201900397. If you can add a comment on the very first steps towards computational MIP design: https://doi.org/10.1002/(SICI)1099-1352(199812)11:1/6<79::AID-JMR394>3.0.CO;2-B.

Response:

Thank you for your comment. Information regarding the approach methods for the selection of templates and monomers has been briefly mentioned in the text (page 2 lines 63 – 65). The sentence “The selection of template and monomer molecules is critical in determining the specificity and properties of the MIP” in line 62, serves as a prefix to convey to the reader that the selection of the appropriate template and monomer can provide good characteristics of MIP.

  1. Introduction (line 48): When listing the different applications of MIP please provide also a brief list of the pro and cons that are related to this technology (all the advantages related to the compatibility with other materials, their stability compared to bio-based systems, the capacity to simulate key-lock enzymatic interactions, the possibility of being redesigned). In respect to this last point you can add this recent example: https://doi.org/10.3390/s19204433.

Response:

Thank you for your comment. We provided additional information regarding the advantages of using MIP for the preparation, analysis, and detection of a compound on lines 58 – 64.

  1. Please improve the figure resolution and make the labels more readable starting from Figure 2.

Response:

We improved the resolution of figures 2 and 3, the resolution of all figures already min 300 dpi.

  1. Section 2 (line 185 on): please consider dividing the paragraph 2.2 in subsections to help the reader going through the text and understanding better the concepts described.

Response:

Thank you for your comment. We have divided section 2.2 into subsections according to the topic of discussion to make it easier for readers to understand.

  1. After section 3, please include a last section 4 for the final remarks and the conclusions. Please provide insights about the future developments of these technology and how the state-of-the-art provided in your review can inspire/contribute to the improve this research field.

Response:

Thank you for your comment. Section 4 has been added on page 18.

Reviewer 4 Report

The manuscript is relevant to the field as it aims at reviewing and comparing  MIP produced with two and one functional monomers. 

However, more work is needed.

Please refer to the following list of major flaws and shortcomings:

1- Attention should be paid to using the correct definitions of MIP, composition (not component) and Sensor. Please refer to the definition/concept accepted in the scientific community.

2. Please specify better what polymer medium means and how this relates to MIP.

3. Avoid personal statements and biased statements. e.g., "Generally, MIP is only synthesised using one type of monomer (mono-functional monomer); however, this type of MIP has limited selectivity and adsorption capacity." There might be a case where MIPs synthesised using one functional monomer works better than aMIp synthesised using two or three functional monomers.

3. Eglish needs moderate to extensive revision. Therefore, rephrasing is also advised throughout the text, e.g. in lines 41-43, 53-54. 96-100, 147-148, 333- 336, 478-484

4. Add references to support your statement, e.g., lines 56-60, 82-83.

5. too much repetition throughout the manuscript. Please avoid repeating the same concept over and over again.

6. Authors should argue better the statement in lines 104 -106, other than giving proper references.

7. Check the numbers you report in the manuscript, e.g. lines 265-266, 272.

8. line 377-379-> Qmax and IF are very close here, so I would not say there are any differences. Also, you must perform/report some statistical analysis to claim differences.

9. Statements reported in lines 398-399 should be stated earlier in the paper.

10. Paragraph discussing MIP sensors needs extensive work as it does not provide a comprehensive background overview, and the comparison between mono and dual monomers MIP is missing.

11. Overall, the manuscript structure is missing and not explained. The manuscript structure should be presented in the introduction to guide the reader. Also, paragraph 2.2 does not flow with the previous one and contains many repetitions. Conclusions are missing.

12. More schemes and figures might also help in improving the manuscript.

Author Response

Please refer to the following list of major flaws and shortcomings:

  1. Attention should be paid to using the correct definitions of MIP, composition (not component) and Sensor. Please refer to the definition/concept accepted in the scientific community.

Response:

Thank you for your comment. We corrected the definition of MIP on page 1 lines 11 – 13.

The correction is below:

A Molecularly imprinted polymer (MIP) is a synthetic polymer that has characteristics such as natural receptors which are able to interact and bind to a specific molecule that is used as a template in the MIP polymerization process.

  1. Please specify better what polymer medium means and how this relates to MIP.

Response:

Thank you for your comment. The word polymer medium refers to a technique or method of preparation that utilizes a polymer as a particle that is capable to absorb the analyte and separate the analyte from the sample matrix. Polymer medium also refers to MIP where MIP is a polymer from the process of combining monomers assisted by a crosslinker to form a polymer that has specific recognition sites. MIP can be used as a component in sample preparation and analysis methods such as solid phase extraction, high-pressure liquid chromatography, and biochemical or electrochemical sensors.

  1. Avoid personal statements and biased statements. e.g., "Generally, MIP is only synthesised using one type of monomer (mono-functional monomer); however, this type of MIP has limited selectivity and adsorption capacity." There might be a case where MIPs synthesised using one functional monomer works better than aMIp synthesised using two or three functional monomers.

Response:

Thank you fo your comment. We changes the sentences that contain personal statements and biased statements. The statement has been adjusted with the results of studies that have been carried out previously.

The correction is below:

Generally, MIPs are only synthesized using one type of monomer (mono-functional monomer); but along with the development of MIP, MIP began to be synthesized using two types of monomers to improve the performance of MIP (page 1 lines 17 – 19).

  1. Eglish needs moderate to extensive revision. Therefore, rephrasing is also advised throughout the text, e.g. in lines 41-43, 53-54. 96-100, 147-148, 333- 336, 478-484

Response:

We did proofreading for the submitted manuscript with proof-reading-service.com with ref. no. 202207-423305. We rephrased some lines mentioned above.

The correction is below

  • Lines 41 – 43: In the MIP synthesis process, the basic compositions are always the same, consisting of functional monomers, crosslinkers, porogenic solvent, and template molecules (now in lines 49 – 51).
  • Lines 53 – 54: The template and monomer molecules usually complement each other (now in lines 65 – 66).
  • Lines 96 – 100: The sample preparation process is an important step in analytical process to purify and concentrate analytes in order to increase sensitivity, selectivity, and accuracy (now in lines 109 – 111).
  • Lines 147 – 148: Epoxy-modified silica nanoparticles were used as solid support in the MIP synthesis by Zhao et al which were then coated with a thin film layer in the imprinting process. (no in lins 167 – 169).
  • Lines 333 – 336: The amount of glycyrrhizic acid that can be quantified if the analyte separation does not use MIP as a sorbent in SPE (MI-SPE) is only about 15.03%. Meanwhile, the amount of glycyrrhizic acid that can be quantified after separation with MI-SPE is 53.63%. (now in lines 392 – 394).
  • Lines 478 – 484: In this method of application, polymerized dual-functional monomer MIPs are used as detection element in sensing device for the identification of molecules of compounds (now in lines 513 – 515).

  1. Add references to support your statement, e.g., lines 56-60, 82-83.

Response:

Thank you for your comment. We added references for page 2 in lines 70 – 74.

  1. Authors should argue better the statement in lines 104 -106, other than giving proper references.

Response:

Thank you for your comment. We added the statement in page 4 lines 119 – 122. The revision is below :

Sorbents that are widely used in SPE, such as silica and C18 have limitations in their selectivity and specificity, resulting in poor separation results. Therefore, a sorbent with better selectivity and specificity is needed, compared with the commercially available sorbent.

  1. Check the numbers you report in the manuscript, e.g. lines 265-266, 272.

Response:

We corrected the numbers in line 265 – 272 (now in lines 297 – 301).

The correction is below :

CIP-MIP synthesized using a dual-functional monomer combination of 2VP and MAA (2VP-MAA-MIP) showed a higher adsorption capacity (Qmax = 2.40 mg/g, IF = 1.66) than 2VP-MIP (Qmax = 1.12 mg/g, IF = 1.51) and MAA-MIP (Qmax = 1.6 mg/g, IF = 0.8), proving that 2VP-MAA-MIP had higher selectivity for CIP (see Qmax and IF values in Table 2).

  1. line 377-379-> Qmax and IF are very close here, so I would not say there are any differences. Also, you must perform/report some statistical analysis to claim differences

Response:

Thank you for your comment. Before, there is an error in the calculation of the data conversion of the Qmax value. We have been fixed it in the revised manuscript. In addition, based on reference journals, Cai et al stated that MIP synthesized using a dual-functional monomer has the possibility to work better than MIP with a single monomer. Then the statement was proven from the results of research conducted also by Cai et al that MIP with dual-functional monomer has a good percent recovery (94.0 – 101.3%).

https://doi.org/10.1016/j.chroma.2021.462009

  1. Statements reported in lines 398-399 should be stated earlier in the paper.

Response:

Thank you for your comment. We stated lines 398 – 399 in the second paragraphs of section 2.2 (lines 229 – 231).

  1. Overall, the manuscript structure is missing and not explained. The manuscript structure should be presented in the introduction to guide the reader. Also, paragraph 2.2 does not flow with the previous one and contains many repetitions. Conclusions are missing.

Response:

Thank you for your comment. We divided section 2.2 into subsections according to the topic of discussion to make it easier for readers to understand.

Round 2

Reviewer 1 Report

Thank you Authors for their response and correction. But there is still some aspect that need modification.

1.      Is the title of subsection 2.1 correct? Dual-Functional Monomer Molecularly Imprinted Solid Phase Extraction Polymerization Techniques? Polymerization techniques are connected with imprinted polymers not with the SPE (SPE is the proces not material synthesized). Authors should consider the title: Polymerization Techniques used in synthesis of Dual-Functional Monomer Molecularly Imprinted Polymers for MI-SPE. In the response to reviewer Authors wrote better explanation of the title.

2.      Lines 135-137: Currently, the polymerization methods that are widely used for the synthesis of dual-functional monomer MI-SPE are precipitation polymerization and bulk polymerization .  – The same comment as above. MI-SPE according to the Authors description is the extraction technique, then we cannot write that we synthesize extraction !! Similar mistake/inaccuracy is present in the whole text.

3.      Lines 232-239 described synthetic process and the subsection 2.2 is connected with the application. That text should be placed in subsection 2.1 connected with the synthesis.

4.      Line 439 consequently  Authors should titled subsection according to the template /analyte to make the text consistent. Not according to the sample type.

5.      The title of subsection 3.1 and 3.2 are not correctly constructed.

6.      The text content some editorial mistakes, e.g. two different ways of Qmax notation, viz Qmax or Qmax.

7.      Probably, addition of the abbreviation list will help the reader to understand and read the text.

Author Response

Reviewer 1

  1. Is the title of subsection 2.1 correct? Dual-Functional Monomer Molecularly Imprinted Solid Phase Extraction Polymerization Techniques? Polymerization techniques are connected with imprinted polymers not with the SPE (SPE is the proces not material synthesized). Authors should consider the title: Polymerization Techniques used in synthesis of Dual-Functional Monomer Molecularly Imprinted Polymers for MI-SPE. In the response to reviewer Authors wrote better explanation of the title.

Response:

Thank you for your comment.

We are not aware of misinterpretation in the title of subsection 2.1. Therefore, we changed the title of subsection 2.1. from “Dual-Functional Monomer MI-SPE Polymerization Techniques” to “Polymerization Techniques of Dual-Functional Monomer MIP as MI-SPE Sorbent” (page 4, lines 114 – 115).

  1. Lines 135-137: Currently, the polymerization methods that are widely used for the synthesis of dual-functional monomer MI-SPE are precipitation polymerization and bulk polymerization .  – The same comment as above. MI-SPE according to the Authors description is the extraction technique, then we cannot write that we synthesize extraction !! Similar mistake/inaccuracy is present in the whole text.

Response:

Thank you for your comment.

We corrected the sentences stating “polymerization method for MI-SPE synthesis”.

The correction is below:

  • Currently, the polymerization methods that are widely used for the synthesis of dual-functional monomer MIP for SPE are precipitation polymerization and bulk polymerization (Figure 1). (page 4, lines 136 – 138).
  • An example of dual-functional monomer MIP as SPE sorbent which was synthesized using the precipitation polymerization method was carried out by Wan et al. (page 5, lines 145 – 147).
  • Nguyen et al. and Zhao et al. also synthesized dual-functional monomer MIP for SPE using the precipitation polymerization method. Nguyen et al. synthesized a dual-functional monomer MIP using ciprofloxacin as a template and a combination of MAA and 2-vinylpiridine as a functional monomer. (page 5, lines 159 – 162).
  • An aprotic polar porogen solvent is suitable for preparing MIP as SPE sorbent by precipitation polymerization. MIP synthesized using a polar protic solvent (methanol) showed a low adsorption capacity (Qmax = 3.23 mg/g) and selectivity (imprinting factor (IF) = 0.67) when compared to MIP synthesized using a polar aprotic solvent (acetonitrile) (Qmax = 11.76 mg/g, IF = 1.34). (page 5, lines 165 – 170).
  • The study by Zhao et al. synthesized a dual-functional MIP to detect polysaccharides in aqueous media by a surface imprinting technique commonly used for macromolecular recognition. (page 5, lines 173 – 175).
  • Based on the Qmax and IF, the use of dual-functional monomers in the synthesis of MIP as SPE sorbent was able to increase the selectivity for the recognition of polysaccharides. (page 5, lines 178 – 180).
  • MIP for SPE sorbent has been widely synthesized for various template compounds in the analysis of food samples, such as coumarin in chamomile tea and cinnamon samples, citrinin in dietary supplements and cereal samples, and melamine and dicyandiamide in diary product samples. (page 8, lines 234 – 237).

  1. Lines 232-239 described synthetic process and the subsection 2.2 is connected with the application. That text should be placed in subsection 2.1 connected with the synthesis.

Response:

Thank you for your comment.

We moved lines 232-239 in subsection 2 to lines 141 - 146 in subsection 1.

  1. Line 439 consequently  Authors should titled subsection according to the template /analyte to make the text consistent. Not according to the sample type.

Response:

Thank you for your comment.

We corrected the title of sub-subsection d in subsection 2.2 (lines 421).

The correction is below:

Dual-Functional Monomer MI-SPE for Starch

  1. The title of subsection 3.1 and 3.2 are not correctly constructed.

Response:

Thank you for your comment.

We corrected the title of subsection 3.1 and 3.2.

  • The title of subsection 3.1: Polymerization Techniques of Dual-Functional Monomer MIP as Recognition Element in Sensor (lines 485 – 486).
  • The title of subsection 3.2: Dual-Functional Monomer MIP Application as Recognition Element in Sensor (lines 533 – 534).

  1. The text content some editorial mistakes, e.g. two different ways of Qmax notation, viz Qmax or Q

Response:

Thank you for your comment.

We corrected all Qmax notation to Qmax.

  1. Probably, addition of the abbreviation list will help the reader to understand and read the text.

Response:

Thank you for your comment.

We added a list of abbreviations to the manuscript before the references section.

Reviewer 4 Report

Thank you for addressing my comments. In my opinion, the paper is suitable for publication.

Author Response

Reviewer 4

Thank you for addressing my comments. In my opinion, the paper is suitable for publication.

Response:

Thank you for your suggestions and comments for improving this manuscript.
